# Utility of the Age Discrepancy between Frailty-Based Biological Age and Expected Life Age in Patients with Urological Cancers

**DOI:** 10.3390/cancers14246229

**Published:** 2022-12-17

**Authors:** Kyo Togashi, Shingo Hatakeyama, Osamu Soma, Kazutaka Okita, Naoki Fujita, Toshikazu Tanaka, Daisuke Noro, Hirotaka Horiguchi, Nozomi Uemura, Takuro Iwane, Teppei Okamoto, Hayato Yamamoto, Takahiro Yoneyama, Yasuhiro Hashimoto, Chikara Ohyama

**Affiliations:** 1Department of Urology, Hirosaki University Graduate School of Medicine, Hirosaki 036-8562, Japan; 2Department of Urology, Aomori Prefectural Central Hospital, Aomori 030-8553, Japan; 3Department of Urology, Mutsu General Hospital, Mutsu 035-8601, Japan; 4Department of Innovation Center for Health Promotion, Hirosaki University Graduate School of Medicine, Hirosaki 036-8562, Japan; 5Research Institute of Health Innovation, Hirosaki University Graduate School of Medicine, Hirosaki 036-8562, Japan; 6Department of Advanced Transplant and Regenerative Medicine, Hirosaki University Graduate School of Medicine, Hirosaki 036-8562, Japan

**Keywords:** frailty, cancer, life expectancy, chronological, biological, survival

## Abstract

**Simple Summary:**

The estimation of biological age is challenging in patients with cancers. We investigated the prognostic significance of biological-expected life age discrepancy using frailty-discriminant scores (FDS) in patients with urological cancers. The frailty-based biological age was 12 years older than the chronological age. The biological-expected life age discrepancy between the frailty-based biological and expected life ages of >5 years was also significantly associated with poor prognosis. A biological-expected life age discrepancy may be a useful tool in estimating frailty and prognosis in patients with urological cancers.

**Abstract:**

**Background**: The estimation of biological age is challenging in patients with cancers. We aimed to investigate frailty-based biological ages using frailty-discriminant scores (FDS) and examined the effect of biological-expected life age discrepancy on the prognosis of patients with urological cancers. **Methods**: We retrospectively evaluated frailty in 1035 patients having urological cancers. Their frailty-based biological age was then defined by the FDS, which is a comprehensive frailty assessment tool, using 1790 noncancer individuals as controls. An expected life age (=chronological age + life expectancy) was subsequently calculated using the 2019 life expectancy table. The primary outcome was the estimation of the biological-expected life age discrepancy between the frailty-based biological age and expected life age in patients with urological cancers. Secondary outcomes were the evaluation of the effect of the biological-expected life age discrepancy on overall survival. **Results**: We included 405, 466, and 164 patients diagnosed with prostate cancer, urothelial carcinoma, and renal cell carcinoma, respectively. The median chronological age, life expectancy, and estimated frailty-based biological age were 71, 17, and 83 years, respectively. The biological-expected life age discrepancy in any urological cancers, localized diseases, and metastatic diseases was −4.8, −6.3, and +0.15 years, respectively. The biological-expected life age discrepancy of >5 years was significantly associated with poor overall survival. **Conclusions**: The biological-expected life age discrepancy between frailty-based biological age and expected life age may be helpful in understanding the role of frailty and patient/doctor conversation.

## 1. Introduction

In an increasingly aging society, the number of elderly patients with urological cancer is also on the increase [1,2,3,4,5]. Thus, there has been a growing interest in measuring the frailty in patients associated with cancers to evaluate the decline in their physical abilities [6,7,8,9,10,11,12]. Since frailty is related to medical costs and prognosis, there is also an urgent need to establish an effective preventive method. However, a highly accurate assessment for frailty is not practical because many items (up to 70) need to be surveyed [13]. Therefore, to improve this problem, we developed and validated a quantitative and comprehensive geriatric assessment tool (frailty-discriminant score: FDS), including 10 items, and reported the utility of this tool for treatment selection and prognosis [14,15]. Furthermore, FDS is active for frailty assessment in both healthy individuals and cancer patients [16]. Accordingly, we hypothesized that FDS in healthy individuals can represent a frailty-based biological age and might be effective in estimating the biological age of patients with urological cancer. Here, we investigated frailty-based biological ages using FDS and examined the effect of biological-expected life age discrepancy on the prognosis of patients with urological cancers.

## 2. Patients and Methods

### 2.1. Design and Ethics Statement

This was a post hoc analysis of a prospective observational study investigating the prevalence of frailty in patients with urological disease (FRAURO study: UMIN000025057). We conducted this study following the Declaration of Helsinki. The ethics committee of the Hirosaki University School of Medicine (2016-1082, 2019-099-1), and all hospitals approved this study. Written consents were not obtained in exchange for public disclosure of the study information (opt-out approach) [17].

### 2.2. Patient Selection and Demographics

We evaluated 1035 patients with urological cancers (prostate cancer: PC, urothelial carcinoma: UC, and renal cell carcinoma: RCC) between August 2013 and May 2021. All patients were treated with a standard of care based on the disease status. The standard of care included surgery, radiotherapy, chemotherapy, immunotherapy, and palliative care following the latest guidelines. We included 1790 noncancer control individuals from the Iwaki Health Promotion Project between 2013 and 2018 for the estimation of frailty-based biological age as a natural process in the general population. The following variables were collected and analyzed: age, sex, body mass index (BMI), handgrip strength, gait speed (the timed get-up-and-go test: TGUG), serum albumin level, estimated glomerular filtration rate (eGFR) [18], hemoglobin level, exhaustion, and depression, including the presence or absence of hypertension, diabetes mellitus, and cardiovascular disease. In patients with cancer, their Eastern Cooperative Oncology Group performance status (ECOG PS), clinical stage, and overall survival (OS) were evaluated. OS was defined from the date of the first evaluation of FDS to the final follow-up or death. Tumor stages and grades were also stratified following the eighth TNM classification edition [19].

### 2.3. Frailty Evaluation and FDS

FDS was evaluated cross-sectionally at the patients’ initial visit for treatment. Frailty was then evaluated before initial treatment in most of the patients, except for patients with castration-resistant prostate cancer and some patients who had tumor recurrence after definitive therapy. FDS included key parameter differences between patients with urological cancers and those without, including assessment of physical capabilities (handgrip weakness and slowed walking speed), biochemical blood tests (serum albumin level, renal function, and hemoglobin level), and self-reported exhaustion and depression. The detailed formula was reported previously [14,15]. The FDS formulas for patients with PC and non-PC are shown in Table 1 and the Appendix A. The major reasons for the separation between patients with prostate cancer vs. patients with other types of urological cancers are gender and frailty characteristics. The FDS was originally developed to separate healthy subjects from urological cancer patients by introducing frailty-related factors [14,15]. UC and RCC include women, while PC includes only men. Furthermore, our previous study [15] suggested that patients with prostate cancer tend to be less frail than those with UC and RCC. This represents that there are many healthy patients who undergo surgery for early-stage prostate cancer in real-world clinical practice. Therefore, we developed different FDS formulas for patients with PC vs. non-PC.

### 2.4. The Estimation of Frailty-Based Biological Age and Life Expectancy

Frailty-based biological ages were defined by FDS from the 1790 community-dwelling adults using nonlinear regression analysis. The life expectancy of patients with urological cancers was calculated using the 2019 life expectancy table [20].

### 2.5. The Definition of Expected Life Age and Age Discrepancy

We defined the expected life age using the formula:**Expected life age (years) = chronological age + life expectancy**

The age discrepancy was defined as the difference between the expected life age and the frailty-based biological age using the formula:**Biological-expected life age discrepancy (years) = frailty-based biological age − expected life age**

### 2.6. Outcomes

The primary outcome was the comparison of biological-expected life age discrepancy between the frailty-based biological and expected life ages in patients with urological cancers. Secondary outcomes included the effect of this biological-expected life age discrepancy on the OS of patients. Additionally, exploratory outcomes were the association of the chronological or biological ages with their geriatric 8 screening score (G8) [21] or their biological-expected life age discrepancy with G8.

### 2.7. Statistical Analysis

The statistical analyses of clinical data were conducted using the GraphPad Prism v.7.00 (GraphPad Software, San Diego, CA, USA), BellCurve for Excel (Social Survey Research Information Co., Ltd., Tokyo, Japan), and R v.4.0.2 (The R Foundation for Statistical Computing, Vienna, Austria). Categorical variables were also compared using Fisher’s exact test or the *χ*^2^ test. Furthermore, quantitative variables were expressed as the median and interquartile range (IQR), while the differences between groups were compared using the Student’s *t*-test (normally distributed data) or the Mann–Whitney *U*-test (nonnormally distributed data). Correlations were evaluated, as well, by linear or nonlinear regression using the coefficient of determination (R^2^). Therefore, R^2^ values of <0.09, 0.09–0.24, 0.25–0.49, >0.50 were considered as none, weak, moderate, and having a strong relationship, respectively [22]. Additionally, a multivariable Cox regression analysis investigated the association of the biological-expected life age discrepancy on OS. *p*-value < 0.05 was considered statistically significant.

## 3. Results

### 3.1. Participants of This Study

The study population and design were summarized in Figure 1. We developed the FDS in Study 1 (training set) and validated in the utility of FDS in Study 2 (validation set) [14,15]. We included noncancer control individuals in this study (*n* = 1790), those were different from the training and validation sets (*n* = 2280 + 310). The baseline characteristics of the noncancer control individuals (*n* = 1790) are shown in Appendix A. Patients with urological cancers (*n* = 1035) included the participants in the training (*n* = 605) and validation (*n* = 258) sets and historical cohort (*n* = 172) (Figure 1). The baseline characteristics of the patients with urological cancers are shown in Table 2. The median follow-up of patients with urological cancers was 32 (IQR: 16–45) months.

### 3.2. Control Individuals and Estimation of Frailty-Based Biological Ages

We estimated frailty-based biological ages by associating their chronological ages and FDS in the noncancer control groups using this formula:**Frailty-based biological age = −0.803 × FDS^2^ + 12.34 × FDS + 59.8**

We observed a “moderate” association between the chronological ages and FDS (R^2^ = 0.332, Figure 2A).

### 3.3. Estimation of Frailty-Based Biological Age in Patients with Urological Cancers

We included 405, 466, and 164 patients diagnosed with PC, UC, and RCC, respectively (Table 3). The distributions of M stage and type of disease were shown in Appendix A. The number of PC, UC, and RCC patients with metastatic disease (M1) included in this study were 94 (23%), 69 (15%), and 59 (36%), respectively (Table 3). From our results, the association between chronological and biological ages in patients with urological cancers was “weak”, with an R^2^ value of 0.134 (Appendix A).

### 3.4. Comparison of Three Types of Ages (Chronological, Expected Life, and Frailty-Based Biological Ages) in Patients with Urological Cancers

Based on the 2019 life expectancy table, we estimated an expected life age in patients with urological cancers (Appendix A). The median chronological age, life expectancy, and estimated frailty-based biological ages were 71, 17, and 83 years, respectively (Figure 2B). The frailty-based biological age was 12 years older than the chronological age. The association between frailty-based biological age and expected life age was “none” with an R^2^ value of 0.07 (Figure 2C). Furthermore, the expected life age (88 years) was significantly older than the frailty-based biological age (83 years, *p* < 0.01). Among the three types of cancers, the frailty-based biological age was significantly older in patients with UC (86 years) than those with PC (81 years) and RCC (82 years) (Figure 2D). The frailty-based biological age was also significantly older in patients with M1 disease than those with M0 disease, whereas this age was not significantly different between patients having M0 and M1 diseases in the chronological age group (71 vs. 71 years) (Figure 2E). Moreover, the expected life age was not significantly different between patients with M0 and M1 diseases (88 vs. 88 years, *p* = 0.85), whereas it was significantly different in the M0 disease group (88 vs. 72 years, *p* < 0.001).

### 3.5. Comparison of Biological-Expected Life Age Discrepancy in Patients with Urological Cancers

The median biological-expected life age discrepancy in the patients diagnosed with any urological cancers, PC, UC, RCC, M0, and M1 diseases was –4.8, –7.0, –2.9, –5.0, –6.3, and +0.15 years, respectively (Figure 2F). The biological-expected life age discrepancy was also significantly different among the PC, UC, and RCC patients (ANOVA, *p* < 0.001). Results showed as well that patients with the M1 disease had significantly higher biological-expected life age discrepancies than those diagnosed with M0 disease (*p* < 0.001).

### 3.6. The Association of the Age Discrepancy with OS

The Kaplan–Meier analysis for OS showed that the biological-expected life age discrepancy of >5 years (*n* = 219) showed poorer survival than that of ≤5 years (Figure 3A). The OS in patients with the M0 disease was significantly poorer in the group with biological-expected life age discrepancy of >5 years than that of others (*p* < 0.001, Figure 3B). The OS in patients with the M1 disease was significantly longer in the group with a biological-expected life age discrepancy of <−10 years than in the group with an age discrepancy of –10 to 5 years (*p* = 0.049), whereas those in the group with a biological-expected life age discrepancy of >5 years had a more significantly shorter OS than that of −10 to 5 years (*p* < 0.001, Figure 3C). The multivariate Cox regression analysis, thus, showed that M1 disease, the biological-expected life age discrepancy of >5 years, ECOG PS, and PC were significantly associated with poor OS (Figure 3D). The hazard ratio (HR) from the background-adjusted multivariable Cox regression analysis showed that the biological-expected life age discrepancy of >5 years was significantly associated with poor OS as well (Appendix A). Furthermore, the percentage of patients with the biological-expected life age discrepancy of >5 years in PC, UC, and RCC patients was 10%, 24%, and 12% among patients with the M0 disease. The percentage of patients with the biological-expected life age discrepancy of >5 years in patients with PC, UC, and RCC was 25%, 46%, and 42% among the M1 disease group (Appendix A). The OS was stratified using the biological-expected life age discrepancy (≤10, −10–5, and >5 years) in patients with PC (Appendix A), UC (Appendix A), and RCC (Appendix A).

### 3.7. The Association between the Age Discrepancy and G8

We evaluated the association of the types of ages with G8 as an exploratory outcome. For this investigation, we included 408 patients having both FDS and G8. The association of their biological ages with G8 was greater than that of their chronological ages (R^2^ = 0.219 vs. 0.040, Figure 4A). Additionally, the G8 of 14 patients corresponded with a biological age of 80 years. Furthermore, the association of the biological-expected life age discrepancy with G8 was weak (R^2^ = 0.183, Figure 4B). The G8 of 14 patients also corresponded with the biological-expected life age discrepancy of –8.63 years. The biological-expected life age discrepancy of 5 years corresponded with the G8 of 8.

## 4. Discussion

Frailty assessment in elderly patients with cancers is important in managing the balance between harms and benefits [23,24,25,26,27,28]. However, a full geriatric assessment in all candidates is time-consuming and is not feasible in clinical practice [10,29,30]. Importantly, the main problem of frailty assessment is the non-intuitiveness of each frailty score, as several frailty assessment tools have identified different frail criteria (a nonfrail, prefrail, or frail) [21,31]. However, meaningful frailty values (such as fried phenotype criteria of ≥3, G8 ≤ 14, or FDS > 2.3) do not easily and intuitively recognize the clinical significances. Accordingly, we tried to convert the frailty score to the scale of age, based on the hypothesis that FDS in healthy individuals can represent a frailty-based biological age [16]. We used “biological-expected life age discrepancy” instead of “biological-chronological age discrepancy” because the meaning of chronological age may vary from region to region. We found that the frailty rapidly increased around 70 years or older in the general population (Figure 2A). According to the formula of frailty-based biological age, we found that the median difference between the chronological and frailty-based biological ages was 12 years (71 vs. 83 years, respectively). This difference was greater in patients with M1 disease (17 years) than those with M0 disease (11 years). The median biological-expected life age discrepancy in patients with M1 disease (+0.15 years; severe frail) was significantly higher than those with M0 disease (−6.3 years; less frail). Moreover, a higher biological-expected life age discrepancy (>5 years; severe frail) had a meaningful impact on OS regardless of the different types of cancer and metastatic status. Therefore, our strategy of converting frailty to biological age may be effective. Further validation studies are, however, necessary.

It should also be recognized that different cancer types have different age discrepancies. The median biological-expected life age discrepancy was the lowest (−7.0 years; less frail) in PC patients compared with UC (−2.9 years; less frail) and RCC (−5.0 years; less frail) patients (Figure 2F). Additionally, the number of patients with a biological-expected life age discrepancy of >5 years (severe frail) was the lowest in PC regardless of metastatic status. These observations suggested that patients with PC were less frail than those with UC and RCC. This observation needs to be validated by other populations.

We also tried to evaluate the association of frailty-based biological ages with G8. Although the relationship was weak (R^2^ = 0.219), we found an improved correlation with frailty-based biological ages than with the chronological age. The G8 of 14, 12, and 10 represented biological ages of 80, 85, and 89, respectively. Additionally, G8 of 14, 12, and 10 represented biological-expected life age discrepancies of –8.5, −4.4, and 0, respectively. Therefore, we were unable to translate our findings for clinical practice, as the conversion of G8 to the biological age was proposed to be useful in understanding the clinical impact of G8.

Finally, several limitations need to be declared. First, the retrospective nature, including the limited sample size, selection bias, mixed cancer types, and clinical stages, were strong limitations of this study. Additionally, results from a single population could not directly apply to other populations. Despite these limitations, our study was the first to report the clinical implication of frailty-based biological age on prognosis in patients with urological cancers. We believe that the strategy of age subtraction from life expectancy can minimize the population difference and the conversion of frailty into a scale of age can be a useful option in enhancing the importance of frailty assessment in clinical practice.

## 5. Conclusions

We observed that the median biological-expected life age discrepancy was −4.8 years. The biological-expected life age discrepancy of >5 years was significantly associated with poor overall survival. The frailty-based biological age may be useful in understanding the role of frailty and the patient/doctor conversation.

## Figures and Tables

**Figure 1 cancers-14-06229-f001:**
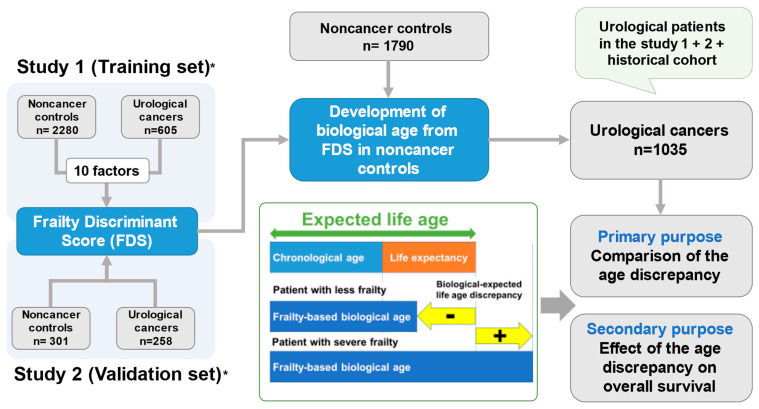
**Study population and design.** We developed the FDS in Study 1 (training set) and validated in the utility of FDS in Study 2 (validation set), which were reported in the previous study (*, refs [7,8]). The definition of the expected life age (years) = [**chronological age + life expectancy**]. The definition of the age discrepancy between the expected life age and frailty-based biological age = [**frailty-based biological age − expected life age**]. A positive value means severe frail, while a negative value means less frail.

**Figure 2 cancers-14-06229-f002:**
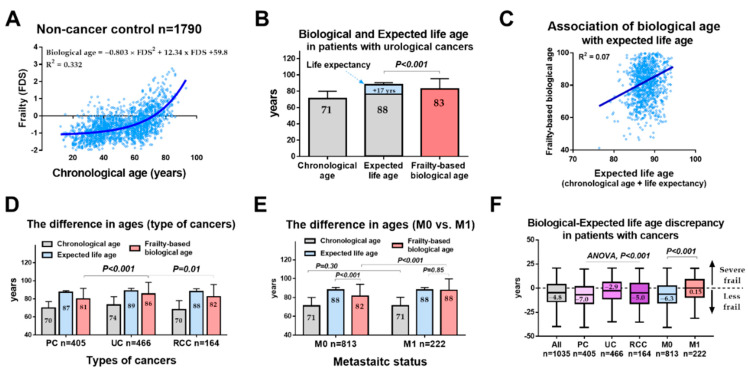
Primary outcomes. (**A**) The estimation of frailty-based biological ages using the frailty-discriminant score (FDS) from noncancer individuals. (**B**) The evaluation of the chronological, frailty-based biological, and expected life ages. (**C**) The association of frailty-based biological age and expected life age in patients with urological cancers. (**D**) The comparison of three types of ages among PC, UC, and RCC patients. (**E**) The comparison between three types of ages between the nonmetastatic (M0) and metastatic (M1) disease groups. (**F**) Estimation of the age discrepancy between the frailty-based biological age and expected life age.

**Figure 3 cancers-14-06229-f003:**
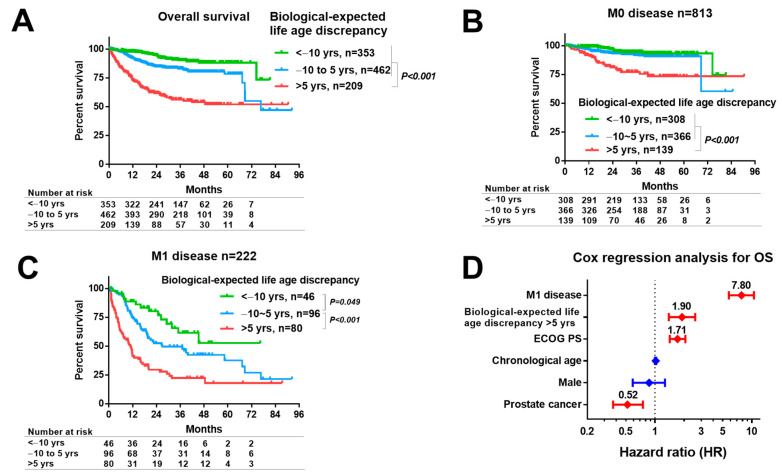
**Secondary outcomes.** (**A**) The OS comparison of age discrepancy among patients with urological cancers. (**B**) The OS comparison of age discrepancy among patients with urological cancers in the nonmetastatic (M0) disease groups. (**C**) The OS comparison of age discrepancy among patients with urological cancers in the metastatic (M1) disease groups. (**D**) Hazard ratio (HR) of multivariable Cox regression analysis to test the association of age discrepancy on OS.

**Figure 4 cancers-14-06229-f004:**
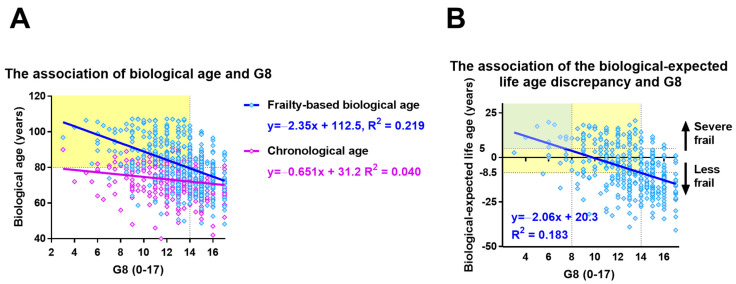
**Exploratory outcomes.** (**A**) Association of the chronological-biological age discrepancy with the geriatric 8 screening score (G8). The yellow part represents G8 ≤ 14 and biological age ≥ 80 years old. (**B**) Association of the biological-expected life age discrepancy with the G8. The yellow part represents G8 ≤ 14 and biological-expected life age ≥ −8.5 years.

**Table 1 cancers-14-06229-t001:** The FDS formulas for patients with PC and non-PC.

Type of Disease	The FDS Formula
**PC**	= 5.6418 + age × 0.0110 + BMI × 0.0267 + handgrip × 0.0094 + TGUG × 0.1960 + exhaustion × −0.0880 + depression × 0.0464 + albumin × −0.5343 + eGFR × 0.0175 + hemoglobin × −0.5204
**Non-PC**	= 6.8698 + age × 0.0053 + sex (male = 1, female = 0) × 1.4794 + BMI × 0.0105 + handgrip × −0.0209 + TGUG × 0.1993 + exhaustion × 0.0876 + depression × 0.2005 + albumin × −0.9037 + eGFR × −0.0112 + hemoglobin × −0.2868

PC: prostate cancer, FDS: frailty-discriminant scores, BMI: body mass index, TGUG: the timed get-up-and-go test.

**Table 2 cancers-14-06229-t002:** Background of participants.

	Urological Cancers
**Number**	1035
**Age, years (IQR)**	71 (66–78)
**Male, n**	880 (85%)
**Body mass index, kg/m^2^ (IQR)**	23 (21–25)
**Handgrip, kg**	30 (22–36)
**Timed get-up-and-go (TGUG), sec.**	9.6 (8.0–12)
**Fatigue (yes), n**	169 (16%)
**Depression (yes), n**	138 (13%)
**Hypertension, n**	485 (47%)
**Diabetes mellitus, n**	209 (20%)
**Cardiovascular disease, n**	154 (15%)
**Albumin, g/dL**	3.9 (3.6–4.3)
**Hemoglobin, g/dL**	12 (11–14)
**eGFR, mL/min/1.73m^2^ (IQR)**	69 (53–84)
**Frailty-discriminant score (FDS)**	2.20 (1.25–3.29)
**ECOG PS, n**	
**0**	856 (83%)
**1**	120 (12%)
**2**	49 (4.7%)
**3 or 4**	10 (1.0%)
**Prostate cancer, n**	405 (39%)
**Urothelial carcinoma, n**	466 (45%)
**Renal cell carcinoma, n**	164 (165)
**Lymph node involvement, n**	90 (8.7%)
**Metastatic disease, n**	222 (21%)
**Deceased, n**	196 (19%)

IQR: interquartile range.

**Table 3 cancers-14-06229-t003:** Detailed information of patients with urological cancers.

**Prostate cancer (PC)**	
**Number, n**	405
**Age, years (IQR)**	70 (65–74)
**Male, n**	405 (100%)
**CRPC, n**	75 (78%)
**M0, n**	336 (77%)
**Surgical treatment**	266 (66%)
**Radiotherapy**	42 (10%)
**nmCRPC**	2 (0.5%)
**M1, n**	69 (23%)
**mCSPC**	22 (5.4%)
**mCRPC**	73 (18%)
**Urothelial carcinoma (UC)**	
**Number, n**	466
**Age, years (IQR)**	74 (67–80)
**Male, n**	355 (76%)
**Bladder cancer, n**	312 (67%)
**M0, n**	280 (60%)
**NMIBC**	83 (18%)
**MIBC**	197 (42%)
**TURBT**	83 (18%)
**Radical cystectomy**	114 (25%)
**Radiotherapy**	79 (17%)
**Best supportive care**	4 (0.9%)
**M1, n**	31 (6.6%)
**Chemotherapy**	25 (5.4%)
**Immunotherapy**	6 (1.3%)
**Upper tract urothelial carcinoma (UTUC), n**	153 (33%)
**M0, n**	117 (25%)
**BCG therapy**	7 (1.5%)
**Radical nephroureterectomy**	81 (17%)
**Radiotherapy**	5 (1.1%)
**Others**	24 (5.1%)
**M1, n**	38 (8.2%)
**Chemotherapy**	33 (7.1%)
**Immunotherapy**	5 (1.1%)
**Renal cell carcinoma (RCC)**	
**Number, n**	164
**Age, years (IQR)**	70 (63–77)
**Male, n**	120 (73%)
**M0, n**	105 (64%)
**Surgical treatment**	99 (60%)
**TKI therapy**	6 (3.7%)
**M1, n**	59 (36%)
**TKI therapy**	43 (26%)
**Immunotherapy**	14 (8.5%)
**Best supportive care**	2 (1.2%)

mCRPC: metastatic castration-resistant prostate cancer, nmCRPC: non-metastatic castration-resistant prostate cancer, mCSPC mCRPC: metastatic castration-sensitive prostate cancer, NMIBC: non-muscle invasive bladder cancer, MIBC: muscle-invasive bladder cancer, TURBT: transurethral resection of bladder tumor, BCG: bacillus Calmette–Guerin, TKI: tyrosine-kinase inhibitor.

## Data Availability

The data presented in this study are available on request from the corresponding author.

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
