# Peer review of "Utility of the Age Discrepancy between Frailty-Based Biological Age and Expected Life Age in Patients with Urological Cancers"

_cancers, 2022, doi:10.3390/cancers14246229_

Round 1
Reviewer 1 Report
The authors retrospectively evaluated the biological age of patients with urological cancers using frailty discriminant scores and than estimated the discrepancy between the latter and the expected life age. They also evaluated the association of age discrepancy to overall survival.
Although I find the results interesting I think the text needs some improvements.
The distribution of patients into such broad groups according to cancer type, that is - prostate cancer, urothelial cancer and renal cancer - is in my opinion not appropriate. All of aforementioned organ-based groups comprise heterogenous types of cancer with different prognosis and different modes of tretment which have an effect on quality of life and then the frailty-based biological age. I would suggest to report the results separately for, for example NMIBC and MIBC.
I did not understand why is the FDS formula different for patients with prostate cancer vs patients with other types of urological cancers or healthy individuals (please offer short explanation in the text regardless if it was given in your previous studies).
Author Response
We sincerely appreciate the kind comments and suggestions of all reviewers and editors. We attach here our revised manuscript, as well as a point-by-point response to the reviewers’ comments.
Comment#1
The distribution of patients into such broad groups according to cancer type, that is - prostate cancer, urothelial cancer, and renal cancer - is in my opinion not appropriate. All of the aforementioned organ-based groups comprise heterogeneous types of cancer with different prognoses and different modes of treatment which have an effect on the quality of life and then the frailty-based biological age. I would suggest to report the results separately for, for example, NMIBC and MIBC.
Response: Thank you for the thoughtful comments. A limitation of our study is that we analyzed a large number of diseases and stages together. While we agree that ideally MIBC and NMIBC should be separated, the major limitation is the sample size. The biggest challenge in measuring flail is time and effort for many patients. We need to analyze by disease and stage if we have a large sample. Our future study needs to address this problem.
Comment#2
I did not understand why is the FDS formula different for patients with prostate cancer vs patients with other types of urological cancers or healthy individuals (please offer a short explanation in the text regardless if it was given in your previous studies).
Response: Thank you for the important comments. The major reasons for the separation between patients with prostate cancer vs patients with other types of urological cancers are gender and frailty characteristics. FDS was originally developed to separate healthy subjects from cancer patients by introducing frailty-related factors. Urothelial and renal cancers include women, while prostate cancer includes only men. Thus, we could not include healthy female individuals as a control to develop the frailty formula for patients with prostate cancer. In this regard, we separated prostate cancer and other types of urological cancers. Furthermore, our previous study suggested that patients with prostate cancer tend to be less frailty than urothelial and renal cell carcinoma. This represents that there are many healthy patients who undergo surgery for early-stage prostate cancer in real-world clinical practice. Therefore, we developed different FDS formulas for patients with prostate cancer vs patients with other types of urological cancers or healthy individuals. This part was added to the methods part.
Page 5 line 111
The major reasons for the separation between patients with prostate cancer vs patients with other types of urological cancers are gender and frailty characteristics. The FDS was originally developed to separate healthy subjects from urological cancer patients by introducing frailty-related factors. UC and RCC include women, while PC includes only men. Furthermore, our previous study suggested that patients with prostate cancer tend to be less frailty than UC and RCC. This represents that there are many healthy patients who undergo surgery for early-stage prostate cancer in real-world clinical practice. Therefore, we developed different FDS formulas for patients with PC vs non-PC.
Reviewer 2 Report
the submitted article highlights an interesting issue. i recommend the authors to include the following articles:
Crocetto F, Buonerba C, Caputo V, Ferro M, Persico F, Trama F, Iliano E, Rapisarda S, Bada M, Facchini G, Verde A, Placido S, Barone B. Urologic malignancies: advances in the analysis and interpretation of clinical findings. Future Sci OA. 2021 Feb 4;7(4):FSO674. doi: 10.2144/fsoa-2020-0210. PMID: 33815820; PMCID: PMC8015670.
I also advise the authors to clarify the objectives of the study in both the introduction and abstract.
Author Response
We sincerely appreciate the kind comments and suggestions of all reviewers and editors. We attach here our revised manuscript, as well as a point-by-point response to the reviewers’ comments.
Comments#1
The submitted article highlights an interesting issue. I recommend the authors include the following articles: Crocetto F, Buonerba C, Caputo V, Ferro M, Persico F, Trama F, Iliano E, Rapisarda S, Bada M, Facchini G, Verde A, Placido S, Barone B. Urologic malignancies: advances in the analysis and interpretation of clinical findings. Future Sci OA. 2021 Feb 4;7(4):FSO674. doi: 10.2144/fsoa-2020-0210.
Response: We agree with the suggestion. We cited this article (Ref. 5).
Comment #2
I also advise the authors to clarify the objectives of the study in both the introduction and abstract.
Response: We agree with the suggestion. We revised the objectives in the abstract.
Background: We aimed to investigate frailty-based biological ages using frailty discriminant scores (FDS) and examined the effect of biological-expected life age discrepancy on the prognosis of patients with urological cancers.